# Determination of Fatty Acid Profile in Processed Fish and Shellfish Foods

**DOI:** 10.3390/foods12132631

**Published:** 2023-07-07

**Authors:** Vincenzo Nava, Vincenzo Lo Turco, Patrizia Licata, Veselina Panayotova, Katya Peycheva, Francesco Fazio, Rossana Rando, Giuseppa Di Bella, Angela Giorgia Potortì

**Affiliations:** 1BIOMORF Department, University of Messina, Polo SS Annunziata, 98168 Messina, Italy; 2Department of Veterinary Sciences, University of Messina, Polo SS Annunziata, 98168 Messina, Italy; 3Department of Chemistry, Medical University of Varna, 9002 Varna, Bulgaria

**Keywords:** seafood processed foods, n-3 PUFA, fatty acids content, health lipid indices

## Abstract

Seafood products are a crucial dietary source of n-3 polyunsaturated fatty acids (n-3 PUFA), which are essential for human health. However, the presence of these n-3 PUFA may be subject to changes related to different processing methods. The aim of this study was to determine the fatty acid composition, focusing on n-3 PUFA, in different processed fish and shellfish products of both EU and non-EU origin. The products were purchased from supermarkets and ethnic food shops in Messina (Italy). Gas chromatography with a flame-ionization detector (GC-FID) was used for analysis. Based on the fatty acid profile, the atherogenicity index (AI), thrombogenicity index (TI), and flesh lipid quality index (FLQ) were determined: 0.13–1.04 (AI), 0.19–0.89 (TI), and 0.41–29.90 (FLQ). The percentages of saturated (SFA), monounsaturated (MUFA), and polyunsaturated (PUFA) fatty acids fell within the following ranges: 13.55–50.48%, 18.91–65.58%, and 13.84–52.73%, respectively. Considering that all samples showed low AI and TI indices and that all processed fish products proved to be a good source of beneficial PUFAs, especially eicosapentaenoic acid (EPA) and docosahexaenoic acid (DHA), their consumption is recommended for humans.

## 1. Introduction

Fish and shellfish products are important for humans due to their many health benefits [1,2]. They contain high-quality protein, lipids, vitamins (D, A, E and B_12_), essential elements (e.g., selenium), and other essential nutrients [3].

Based on their lipid content, fish can be classified into three categories: lean fish (2–5 percent fat, e.g., cod and sole), medium-fat fish (5–6 percent fat, e.g., hake and sea bass), and fatty fish (6–25 percent fat, e.g., anchovies, herring, sardines, mackerel, tuna, and salmon) [1]. Depending on factors such as species, age, sex, and season, the fatty acid composition of the lipid fraction varies [1].

The importance of fish and shellfish products lies in their significant contribution as a dietary source of n-3 polyunsaturated fatty acids (n-3 PUFAs) [4]. Among the n-3 PUFAs, eicosapentaenoic acid (EPA, C20:5 n-3) and docosahexaenoic acid (DHA, C22:6 n-3) are the most beneficial [5]. Several studies have shown a close correlation between the consumption of fish species and human health benefits. Indeed, these fatty acids play a crucial role in the synthesis of certain eicosanoids (e.g., prostaglandins, thromboxanes, and leukotrienes) and possess important antithrombotic properties. They also reduce the risk of coronary and cardiovascular disease, and prevent cancer, diabetes, and other inflammatory and autoimmune diseases [6,7]. In addition, n-3 PUFAs are essential throughout all stages of human development, starting from conception, and contribute to normal neurological development in children [1]. They also reduce the risk of dementia disorders and Alzheimer’s symptoms in the elderly [5,8].

EPA and DHA are essential for the normal functioning of the human body. While humans have the ability to synthesize LC-PUFAs, the activity of the ALA desaturase enzyme complex varies and is generally relatively low. Therefore, dietary supplementation with n-3 PUFA is required [9]. This should primarily be achieved by eating fish products. According to the American Heart Association guidelines, eating fish at least twice a week provides approximately 250 mg/day of EPA + DHA [4,8]. Studies conducted by the EFSA Panel on Nutrition, Novel Foods, and Food Allergens (NDA EFSA Panel) have found that consumption of 1–2 servings of seafood products per week, and up to 3–4 servings per week for pregnant women, is associated with functional benefits related to neurodevelopment in children and a lower risk of mortality from coronary heart disease in adults [10].

A balanced n-6/n-3 PUFA ratio in the diet is of fundamental importance. In fact, an unbalanced ratio can lead to chronic diseases due to the competition between n-3 and n-6 PUFA for the same enzymes [11]. For example, some studies [7] recommend a PUFA n-6/n-3 ratio of 4:1 for the prevention of coronary artery disease due to the different roles played by n-3 and n-6: the former contribute to the anti-inflammatory and anti-thrombotic process, while the latter contribute to the pro-inflammatory process. In this regard, the European Scientific Committee on Food and the French Agency for Food Safety have set daily intakes of n-6 PUFA and n-3 PUFA for adults of 6 g/day and 2 g/day, respectively [12].

The consumption of fish products is therefore essential to increase the intake of n-3 LC PUFA. However, due to the frenetic lifestyles in the modern world, more and more processed seafood is being consumed. This term refers to all foods that have undergone manufacturing processes to extend their shelf life or modify their sensory characteristics [13].

The application of transformation processes to food products can have advantages and disadvantages. On the one hand, these processes not only prolong the shelf life of a product and maintain its quality [14], but they also allow for the elimination of most microorganisms [15]. On the other hand, they can play a key role in nutrient retention. For example, heat treatment can cause undesirable changes, such as loss of the nutritional food value. In fact, cooking is a critical step in preserving fish EPA and DHA, as both are highly sensitive to oxidation [16].

To mitigate the adverse effects associated with processing, it is often necessary to incorporate other functional foods to increase the amounts of fatty acids in the starting samples [17,18,19,20]. This choice is based on consumer demand for foods that can provide health benefits, such as improvements in body function or a reduction in the risk of certain diseases. These benefits are linked to the different types of fish processing. For example, Kitts et al. [21] demonstrated how traditional smoked processing provides benefits in terms of reducing lipid oxidation, preserving essential fatty acids, and potentially significant reducing Listeria contamination.

In this research, several processed fish products purchased in supermarkets and ethnic food shops in Messina (Italy) were analyzed. The aim was to determine the fatty acid composition of the samples under examination, with particular attention to the long-chain n-3 PUFA. In addition, lipid quality indices, such as the atherogenicity index (AI), thrombogenicity index (TI), and flesh lipid quality (FLQ) index, were assessed to evaluate the health potential of the lipids in the analyzed processed products.

## 2. Materials and Methods

### 2.1. Samples

In this study, a total of 47 samples of processed fish (tuna, mackerel, sardines, salmon, shrimps, and crabs) from different brands were purchased in supermarkets and ethnic food shops in Messina between September and October 2022 (Table 1). These processed fish products under investigation were the same as those analyzed in a study previously conducted by Nava et al. [13], where their mercury content was determined.

The fatty acid content of each sample was determined before the expiry date of the product.

### 2.2. Chemicals and Reagents

Analytical grade reagents and chemicals were purchased from Merck (Darmstadt, Germany) and used in the study. For fatty acid analysis, a mixture of fatty acid methyl esters (Supelco 37-Component FAME Mix) purchased from Sigma-Aldrich (Darmstadt, Germany), was employed. The other fatty acids, market with an asterisk (*) in Table 2, were identified by comparing them with a well-characterized fish oil (cod liver oil from Supelco).

### 2.3. Lipids Extraction and Preparation of FAMEs

The extraction of the lipid fraction was carried out using the Folch method [22], with some modifications. The entire contents of each product (for cans) were homogenized, and 4 g of the homogenates were weighed into 50 mL tubes. Folch’s solution (chloroform:methanol, 2:1, *v*/*v*) was added to the tubes, followed by the addition of a 0.73% NaCl solution. Subsequently, the samples were vortexed for 1–2 min and then centrifuged at 3500 rpm for 15 min s at 4 °C. The lipid fraction (bottom layer) was recovered in a previously weighed flask and dried using a rotating evaporator (Heidolph Instruments GmbH & Co., Schwabach, Germany). The total lipid contents (g/100 g) were gravimetrically determined.

Fatty acid methyl esters (FAME) were obtained through transmethylation (hot esterification) of the lipid fractions in the analyzed samples. This was achieved by adding a methanol:sulphuric acid mixture (9:1, *v*/*v*) according to the ISO 5509 2000 method [23]. The mixture was heated in an oven at 100 °C for one hour. To protect polyunsaturated fatty acids (PUFA) from high temperatures, butylated hydroxytoluene (BHT) was used as an antioxidant. The supernatant was diluted with n-hexane after collection.

### 2.4. GC-FID Analysis

The fatty acid composition of the investigated samples was analyzed using a gas chromatograph equipped with a split/splitless injector and a flame ionization detector (GC-FID, Dani Master GC, Dani Instrument, Milan, Italy). The instrument was equipped with a ZB-Wax column (Phenomenex, Torrance, CA, USA) with a length of 30 m, an internal diameter of 0.25 mm, and a film thickness of 0.25 µm.

The operating conditions were as follows: the column oven temperature ranged from 50 °C (hold time 2 min) to 240 °C (hold time 15 min) with a heating rate of 3 °C/min. The injector and detector temperatures were set at 240 °C. Helium gas was used as the carrier gas, flowing at a linear velocity of 30 cm/s (constant). The injection volume was 1 µL, with a split ratio of 1:50.

Clarity Chromatography v4.0.2 software (DataApex, Prague, Czech Republic) was used for data acquisition and management. Each sample was analyzed in triplicate, along with the analytical blanks. FAMEs of nutritional interest were identified by comparing their retention times with reference compounds present in two mixtures: Supelco 37-component FAME Mix and Supelco-cod liver oil. The following compounds were present in the first mixture: C4:0, C6:0, C8:0, C10:0, C11:0; C12:0, C13:0, C14:0, C14:1, C15:0, C15:1, C16:0, C16:1, C17:0, C17:1, C18:0, C18:1 n-9t, C18:1 n-9c, C18:2 n-6t, C18:2 n-6c, C20:0, C18:3 n-6, C18:3 n-3, C20:1, C21:0, C20:2, C20:3 n-6, C22:0, C20:3 n-3, C22:1 n-9, C20:4 n-6, C23:0, C22:2, C20:5 n-3, C24:0, C24:1, and C22:6 n-3. The remaining compounds were found in the second mixture.

### 2.5. Atherogenicity Index (AI) and Thrombogenicity Index (TI)

To assess and provide information on the health and nutritional potential of the lipids in the analyzed samples, the atherogenicity index (AI) and the thrombogenicity index (TI) were evaluated [24,25].

The atherogenicity index expresses the ratio between the sum of the main saturated fatty acids and the sum of the main classes of unsaturated fatty acids. The former is considered pro-atherogenic as it promotes the adhesion of lipids to the cells of the immunocirculatory system. The latter are considered anti-atherogenic, as they inhibit plaque aggregation and reduce the levels of esterified fatty acids, cholesterol, and phospholipids, thus playing a preventive role in the development of coronary micro- and macro-pathologies [24,26,27].

The following equation was used to calculate the AI [24,25]:AI = [C12:0 + (4 × C14:0) + C16:0]/[Σ n-6 PUFA + Σ MUFA + Σ n-3 PUFA](1)

The thrombogenicity index expresses the propensity for clot formation in blood vessels and is obtained by calculating the ratio of pro-thrombogenetic fatty acids (SFA) to anti-thrombogenetic fatty acids (MUFA, n-6 PUFA e n-3 PUFA) [16,24].

The TI was calculated using the following equation:TI = [C14:0 + C16:0 + C18:0]/[0.5 × Σ n-6 PUFA + 0.5 × Σ MUFA + 3 × Σ n-3 PUFA + (n-3 PUFA/n-6 PUFA)](2)

### 2.6. Flesh–Lipid Quality Index

The flesh–lipid quality (FLQ) index quantifies the percentage correlation between the main n-3 PUFA (EPA + DHA) and the total fatty acids. Higher values of this index indicate a higher quality of the dietary lipid source [28].

The expression used to calculate FLQ index is given below:FLQ = 100 × [EPA + DHA]/[% of total fatty acids](3)

### 2.7. Statistical Analysis

The significance of the results was analyzed using the SPSS 13.0 software package for Windows (SPSS Inc., Chicago, IL, USA). The non-parametric Kruskall–Wallis test was used to compare the means of the groups of measured data.

## 3. Results and Discussion

The fatty acid composition of the analyzed samples is shown in Table 2, which lists only those fatty acids with a content exceeding 1% in at least one of the products.

In general, the most abundant fatty acids found in all samples were DHA (C22:6 n-3), EPA (C20:5 n-3), palmitic acid (C16:0), stearic acid (C18:0), oleic acid (C18:1 n-9) and linoleic acid (C18:2n-6). This was a reliable result, considering that fish products are good sources of n-3 PUFAs, as well as the different type of vegetable oils added during the transformation process.

Moreover, these results are comparable with those obtained by Mesa et al. [29] in their work on fresh fish, shrimps, and shellfish samples, Grazina et al. [30] on samples of wild and farmed salmon, and by Maldonado-Pereira et al. [31] in their investigation of ultra-processed foods. However, the fatty acid composition proved to be highly variable and strongly dependent on the different types of products investigated.

The percentage of saturated fatty acids (SFA) ranged from 13.55 ± 1.08% in canned tuna pâté samples (F9) to 50.48 ± 1.77% in dried shrimps (F11).

Palmitic acid (C16:0) was the most abundant of the saturated fatty acids, with its percentages decreasing in the following order: F6, F11, F4, F12, F1, F10, F8, F7, F3, F2, F5, and F9. Li et al. [32], in their study on freshwater and marine fish and shrimp from China, also observed a correlation between the percentage of C16:0 and the type of product analyzed. For example, their study indicated a C16:0 percentage close to 30% in mackerel samples, which is comparable to the values found in two mackerel samples (F1 and F4) in this study, but higher than those observed in F2 and F3. Regarding the shrimp samples (F11 and F12), the percentages were comparable to those obtained by Li et al. [32] for different types of fresh shrimp. This shows that the product processing did not significantly affect the percentage of C16:0. Latyshev et al. [33] published a study on the determination of lipids and fatty acids in fresh crabs from the Pacific Northwest, showing varying percentages of C16:0 across different species of crab. However, their results were consistently lower than ours. Dried and canned sardine samples (F5 and F6) showed different percentages of C16:0 according to the different processing procedures of the raw product. However, these results were comparable to those reported by Tarley et al. [34] for canned sardines. In relation to canned tuna pâté samples (F9), Mesias et al. [35], who determined the fatty acid profile in tuna samples after different sterilization treatments, showed a C16:0 concentration of 5%, slightly lower than the results of this study. Finally, the salmon samples (F7 and F8) also had a C16:0 percentage comparable to that found in the study by Grazina et al. [30].

Stearic acid (C18:0) had the highest percentage in F11, followed by F10, F4, F6, F1, F3, F5, F12, F8, F2, F7, and F9. Among the mackerel samples, only F2 showed a lower percentage of this fatty acid compared to the findings reported by Li et al. [32], whereas the samples of shrimp had comparable results. Similar to C16:0, the crab samples were characterized by a higher percentage of C18:0 compared to a study conducted by Latyshev et al. [33]. The remaining samples had C18:0 contents comparable to the respective studies used for comparison [30,34,35].

Myristic acid (C14:0) was found in moderate percentages, ranging from 0.07 ± 0.01% (F9) to 6.26 ± 0.25% (F1). Tarley et al. [34] reported slightly higher C14:0 content in canned sardine samples compared to the samples analyzed in this study, while Latyshev et al. [33], Mesias et al. [35], and Grazina et al. [30] found values comparable to those observed in this survey for samples of crab, canned tuna, and salmon. Furthermore, in comparison to the study of Li et al. [32], the shrimp samples had significantly higher C14:0 content, whereas the F3 mackerel samples had significantly lower content. The lower content of C14:0 was related to the remaining SFAs analyzed.

Lauric acid (C12:0) content was lowest (0.01%) in F3 and F9 and highest (0.23–0.26%) in shrimp products (F11 and F12). In general, shrimps do not contain high concentrations of C12:0. For example, in a study by Yerlikaya et al. [36], lauric acid had the lowest percentages among the saturated fatty acids (between 0.03–1.48%), comparable to the samples analyzed in this study. The increase in the C12:0 content could be due to the presence of coconut oil, one of the richest sources of this fatty acid. Coconut oil is commonly used in South American cuisine and is known to improve the growth performance of fish and their resistance to *Aeromonas hydrophila* [37]. Therefore, it can be assumed that coconut oil may have an influence on shrimp, either through its inclusion in their diet or through contact during one of the production stages. This may have resulted in a slight increase in the C12:0 content.

Canned sardine (F5) and grilled mackerel fillets in olive oil (F3) samples exhibited significantly higher percentages of monounsaturated fatty acids (MUFA). The most abundant MUFAs present were oleic acid (C18:1 n-9, 11.06 ± 0.88–60.11 ± 2.01%), cis-vaccenic acid (C18:1 n-7, 0.58 ± 0.13–5.68 ± 0.38%), palmitoleic acid (C16:1 n-7, 0.15 ± 0.01–8.87 ± 1.01%), and *cis*-11-eicosenoic acid (C20:1 n-9, 0.25 ± 0.10–4.63 ± 0.48%). This result is in line with previous studies used for comparison [30,32,33,34,35].

The composition of polyunsaturated fatty acids (PUFA) also varied depending on the type of processed product analyzed. The significantly highest percentage (52.73 ± 2.88%) was obtained for the canned tuna pâté sample (F9), with linoleic acid (C18:2 n-6, 50.85 ± 3.55%) as the main contributor. The other samples exhibited a range of PUFA content between 13.84 ± 0.66% and 35.46 ± 2.10%. The most prominent PUFAs, other than linoleic acid, were eicosapentaenoic acid (EPA, C20:5 n-3) and docosahexanoic acid (DHA, C22:6 n-3). Significantly higher percentages of EPA (12.08 ± 0.80% and 9.70 ± 0.51%) were found in canned horse mackerel (F1) and natural shrimp (F12), respectively, while dried sardines (F6) exhibited significantly higher levels of DHA (25.09 ± 1.55%).

DHA was found in higher amounts than EPA in seven of the twelve fish categories analyzed (Table 2), resulting in a DHA/EPA ratio ranging from 6.34 (a significantly higher value) in dried sardines (F6) to 0.39 in canned sardines (F5). The consumption of fish is of paramount importance, as it provides significant amounts of n-3 PUFA, which are beneficial for various consumer groups, particularly those with higher demands (e.g., children and pregnant women) [38,39,40]. It is also important to encourage the consumption of n-3 PUFA-rich products to balance the n-6/n-3 PUFA ratio. The DHA/EPA ratios can be compared with those reported in a study conducted by Mesa et al. [29], which examined fresh fish, shrimp, and shellfish samples. The comparison showed that the DHA/EPA ratios in salmon, tuna, and shrimp were similar, whereas in mackerel and sardines, the DHA/EPA ratios we obtained were either lower or equal.

High percentages of ∑ n-3 PUFA were found in samples F1, F4, F6, F7, F8, and F12.

In the analyzed samples, the amounts of n-3 PUFA were always higher than those of n-6 PUFA, except in the samples of canned tuna pâté (F9), where ∑ n-6 was higher than ∑ n-3. For this sample type, the n-6/n-3 PUFA ratio was significantly higher (31.48 ± 2.01%) compared to the other samples. The remaining categories exhibited low n-6/n-3 PUFA ratios (ranging between 0.08 for natural Alaskan salmon and 0.91 for canned sardines) (Table 2). This was an expected result, given the higher abundance of n-3 PUFA compared to n-6 PUFA in marine organisms, as demonstrated by other studies in the literature [41]. It is essential to note that the n-6/n-3 ratio serves as an index of the nutritional quality of marine lipids. A low ratio in the daily diet, usually less than one, is associated with a reduced risk of certain diseases [42,43].

Another parameter used to assess the quality of dietary lipids is the ratio of polyunsaturated fatty acids to saturated fatty acids (PUFA/SFA). Considering that, in 2004, the UK Ministry of Health recommended a minimum PUFA/SFA ratio of 0.45 [44], all the analyzed samples appeared to retain a PUFA/SFA ratio above the recommended level, even after processing. In particular, the canned tuna pâté samples (F9) showed a significantly higher value (3.89 ± 0.87) (Table 2).

The sum of EPA and DHA is also an important indicator of the nutritional quality of lipids in fish products. In a study conducted by Larsen et al. [45] on the effects of cooking methods on the fatty acid profile of New Zealand King salmon (*Oncorhynchus tshawytscha*), the sum of EPA and DHA in all the analyzed products was lower than that found in the salmon samples in this study. This further confirms the influence of processing treatments as a possible cause of the decrease in these fatty acids [45].

Concerning the canned mackerel samples (F1, F2, F3 and F4), the obtained results were compared with the study conducted by Merdzhanova et al. [46] on the determination of fatty acid composition of raw and cooked Black Sea horse mackerel samples. A significant difference in lipid composition was observed from the comparison. The F1 and F4 samples were characterized by a higher presence of C18:0, C18:1 n-9, and C20:5 n-3, and a lower percentage of C16:0, C18:2 n-6, and C22:6 n-3. For the F2 and F3 samples, C18:0 and C18:2 n-6 were comparable, C18:1 n-9 and C20:5 n-3 were higher, and C16:0 and C22:6 n-3 were lower. This discordance could be primarily due to the different origin of the analyzed products, which is linked to other factors and to the canning process. The higher oleic acid content found in the samples is clearly explained by the presence of olive oil.

The fatty acid profiles obtained for the salmon samples (F7 and F8) agreed with a previous study conducted by Blanchet et al. [47] on samples of wild and farmed Atlantic salmon. According to that study, wild salmon is richer in n-3 PUFA and saturated fatty acids (SFA). These findings perfectly align with the results of our study.

In the canned tuna pâté samples (F9), as mentioned above, the order of abundance was PUFA > MUFA > SFA. The percentage of saturated fatty acids was lower than that reported in the literature on fresh tuna [48]. This trend was observed in a study on the change in fatty acid composition after canning [48], where the amounts of C14:0, C16:0, and C18:0 were lower in canned tuna and sardines compared to fresh tuna and sardine products. As mentioned by Selmi et al. [48], the difference in fatty acid composition between canned and raw products is explained by the addition of other ingredients to the fish, such as olive oil or tomato sauce, which increase certain fatty acids (e.g., C18:2 n-6 and C18:1 n-9) and decrease others (e.g., SFAs). This effect could also be due to the cooking process prior to canning, which reduces the percentage of SFAs [48]. The unsaturated fatty acid content of the canned tuna samples was higher than that of the fresh tuna samples. The high levels of linoleic and oleic acids can be attributed to the presence of 20% corn seed oil (as indicated in the product label) used to manufacture the tuna paté. Other declared ingredients in the tuna paté, such as green olives, capers, and oregano, may have influenced its FA composition. Thus, as expected, the incorporation of other ingredients, especially different varieties of oils, in tuna increases the amount of MUFA and PUFA compared to fresh samples. This result obtained in our study agrees with other studies in literature where the addition of olive oil influences the fatty acid composition [48]. In addition, compared to the results reported by Mesias et al. [35], who determined the fatty acid profile in tuna samples after different sterilization treatments, our F9 samples showed a higher percentage of PUFA (52.73% vs. 31.6%), a lower percentage of MUFA (30.56% vs 56.3%), and a higher content of SFA (13.55% vs. 9.0%). In line with expectations, this demonstrates the important influence of different food processing on the fatty acid composition [35].

Regarding the samples of canned (F5) and dried (F6) sardines, the different processing procedures resulted in changes in the fatty acid composition [49]. However, other parameters such as species, diet, season, reproductive status, size, and environmental conditions also influence the proximate and fatty acid composition of fish species [50,51]. In the literature, some studies have shown higher concentrations of certain fatty acids in fresh sardines. For example, research conducted by Moussa et al. [52], which analyzed the nutritional value and fatty acid composition of various cooked fish, including sardines, using atherogenicity and thrombogenicity indices, showed a higher percentage of saturated fatty acids, except for sardines cooked by boiling. In particular, the amounts of C14:0, C15:0, C16:0, and C18:0 were significantly higher than in the canned sardines analyzed in the present work. This clear difference could be due to the different processing methods, as the same trend was not observed for dried sardines. On the other hand, the canned sardines had higher levels of oleic acid (60.11 ± 2.01% vs. 5.3 ± 0.15-24.0 ± 0.54%) and linoleic acid (5.71 ± 0.80% vs. 1.57 ± 0.08%), expect for sardines prepared by boiling (51.7 ± 0.25%), due to the presence of olive oil, compared to the study by Moussa et al. [52]. Furthermore, while the EPA content was comparable for F5 and F6, the DHA content was significantly higher in dried sardines (F6) than in canned sardines (F5). The values reported by Moussa et al. were different depending on the cooking process: the EPA content was higher than in our products for sardines cooked in the microwave (5.3 ± 0.36% vs. 3.48 ± 0.62–3.96 ± 0.22%), but lower for grilled sardines (2.7 ± 0.25% vs. 3.48 ± 0.62–3.96 ± 0.22%), while the DHA content was always lower than in our processed sardines [52].

The fatty acid composition of the crab samples (F10) showed a higher percentage of SFA (30.71 ± 1.46%), followed by MUFA (29.58 ± 1.23%) and PUFA (21.33 ± 1.59%). This result was not in agreement with the study conducted by Latyshev et al. [33] on the determination of lipids and fatty acids in fresh crabs from the Pacific Northwest, and the study by Cherif et al. [53] on the determination of the fatty acid composition of green crabs from the Tunisian region. In both of these studies, the order of abundance of the different fatty acid classes was opposite to that observed in our research. Again, these discrepancies in fatty acid content could likely be caused by different product processing techniques [54,55], considering the well-established knowledge that marine invertebrate lipids are rich in PUFA [33]. In addition, it is very likely that species-specific differences, various feedstuffs, and different natural environmental temperatures within crab habitats can also influence the FA composition of the raw materials used for processing [56,57,58].

Finally, regarding processed shrimp products (F11 and F12), the trend differed according to the type of processing. For F12, higher percentages of PUFA (31.08 ± 1.50%) were obtained, followed by MUFA (35.01 ± 0.63%) and SFA (29.66 ± 1.33%). On the other hand, F11, which was subjected to a drying process, showed an opposite composition: SFA (50.48 ± 1.77%), MUFA (29.40 ± 0.33%), and PUFA (15.16 ± 1.21%). Furthermore, when comparing these results with the study conducted by Li et al. [32] on the fatty acid composition of fresh Chinese shrimps, the percentages obtained for F12 were comparable, whereas those of F11 were different. The latter showed higher percentages of SFAs (50.48% vs. 30.01%), lower percentages of PUFAs (15.16% vs. 40.06%), and comparable percentages of MUFAs (29.40% vs. 24.76%) compared with the study by Li et al. [32].

Table 2 shows a comparison of the atherogenicity index (AI), thrombogenicity index (TI), and flesh lipid quality (FLQ) values found in the analyzed samples. In general, high AI and TI values can lead to platelet aggregation and thrombus formation, whereas values less than one are considered low and therefore beneficial to human health [24]. In this study, only the category of dried shrimp (F11) showed an average AI value slightly above one (1.04 ± 0.28), whereas TI values were always below one. The obtained results are comparable with those reported by Luczynska et al. [2] in their study on several freshwater and marine species in the Polish market, and with those reported by Di Bella et al. [6] concerning anchovy samples. Furthermore, the AI index results were lower than those reported by Aberoumand and Baesi [59] for samples of processed and raw fish fillets. In comparison with the study conducted by Moussa et al. [52] on different types of raw and cooked fish, the results were similar for salmon but lower for tuna, mackerel, and sardines.

The FLQ values ranged from 0.41 in canned tuna pâté (F9) to 29.90 in dried sardines (F6) (Table 2). As expected, the FLQ index of the canned tuna pâté samples was significantly lower compared to the other samples, due to the influence of the olive oil composition on the fatty acid profile. For several of the analyzed products (F1, F2, F4, F6, F7, F8, F9, F10, F11, and F12), these values were comparable to those reported by Luczynska et al. [2], while the remaining samples (F3 and F5) showed significantly lower FLQ values. Higher values of this index indicate a higher quality of the dietary lipid source [24,60].

Table 3 reports the percentage of fatty acids (∑ SFA, ∑ MUFA, and ∑ PUFA) and quality index values in fresh fish samples documented in the literature.

## 4. Conclusions

Most of the analyzed samples proved to be a good source of fatty acids, with the highest abundances observed in DHA (C22:6 n-3), EPA (C20:5 n-3), palmitic acid (C16:0), stearic acid (C18:0), oleic acid (C18:1 n-9), and linoleic acid (C18:2 n-6). The lipid quality of the processed products was assessed using the AI, TI, and FLQ indices. The results demonstrated the good quality of the lipid fraction of the analyzed samples, with the atherogenicity and thrombogenicity indices consistently remaining below 1, which poses no risk to human health. In addition, almost all the samples analyzed were also found to be good sources of beneficial PUFAs, in particular eicosapentaenoic acid (EPA) and docosahexaenoic acid (DHA). With the exception of canned tuna pâté (F9), the percentage of n-3 PUFA was significantly higher than that of n-6 PUFA. In conclusion, the consumption of these processed products is recommended for humans.

## Figures and Tables

**Table 1 foods-12-02631-t001:** Information on processed fish and shellfish products (* data not available). Ingredients are listed in the order given on the label.

Sample Code	Sample	Number of Samples	Species	Country or Origin	Lot	Manufacture Data	Processing Condition	Expired Data	Ingredients
F1	Canned horse mackerel	3	*Trachurus* *murphyi*	Chile	*	*	Canning	24 May 2025	mackerel, water, salt
F2	Canned mackerel fillets	5	***	Portugal	*	*	Canning	11 November 2026	mackerel fillets, olive oil, salt
F3	Grilled mackerel fillets in olive oil	5	*Scomber* *japonicus*	Portugal	L301U-B12	*	Grilling and canning	October 2024	mackerel (87%), organic extra virgin olive oil (10%), salt (3%)
F4	Natural grilled mackerel fillets	5	*Scomber* *japonicus*	Portugal	L309U-B2	*	Grilling and canning	November 2024	mackerel (87%), water, salt
F5	Canned sardines	4	*Sardina* *pilchardus*	Morocco	LKX149L/S	07 February 2019	Canning	29 May 2026	Sardines, olive oil (29%), salt
F6	Dried sardines	5	*Sardinella* *aurita*	Argentina	*	*	Drying	*	*
F7	Canned pink Salmon	3	***	USA	3028163	*	Canning	31 December 2022	Salmon, water, salt
F8	Natural Alaskan salmon	5	*Oncorhynchus nerka*	Denmark	L13221-3	*	Canning	31 December 2025	Salmon, water, salt
F9	Canned tuna pâté	3	*Euthynnus (Katsuwonus) pelamis*	Italy	F048-T1	*	Canning	February 2025	tuna (37%), water, corn seed oil (20%), green olives, soya protein, modified maize starch, capers (1%), salt, oregano, natural flavoring: nutmeg with other natural flavorings
F10	Canned crab meat	3	***	Indonesia	CMCRCI064T	*	Canning	7 February 2023	Crab meat, water, salt, sugar, acidifier (E330), sodium metabisulphite, E385
F11	Dried shrimp	3	*	Argentina	*	*	Drying	*	*
F12	Natural shrimp	3	***	Italy	L20058	*	*	27 February 2023	peeled and pre-cooked shrimps, water, salt, sugar, citric acid. May contain traces of shellfish and fish
Tot		47							

**Table 2 foods-12-02631-t002:** Total fat (g/100 g) and fatty acid composition (% of total FA) of processed fish foods. Fatty acids indicated with an asterisk were identified using fish oil (cod liver oil from Supelco) as a reference. See Table 1 for the correspondence between code and sample type. Values with different superscript letters (A, B or C) are significantly different at the 0.05 level.

	F1	F2	F3	F4	F5	F6	F7	F8	F9	F10	F11	F12
Fat	8.62	25.65	10.17	5.79	44.81	1.69	3.60	2.08	11.84	0.64	3.48	1.13
C12:0	0.09 ± 0.01	0.03 ± 0.00	0.01 ± 0.00	0.07 ± 0.02	0.03 ± 0.00	0.13 ± 0.02	0.09 ± 0.01	0.05 ± 0.01	0.01 ± 0.00	0.18 ± 0.02	0.23 ± 0.03	0.26 ± 0.05
C14:0	6.26 ± 0.25	3.51 ± 0.37	0.89 ± 0.20	4.21 ± 0.61	1.43 ± 0.23	3.77 ± 0.51	4.91 ± 0.27	3.74 ± 0.44	0.07 ± 0.01	1.70 ± 0.31	4.73 ± 0.22	2.90 ± 0.30
C15:0	0.47 ± 0.05	0.26 ± 0.03	0.23 ± 0.07	1.07 ± 0.33	0.09 ± 0.01	1.03 ± 0.41	0.70 ± 0.17	0.55 ± 0.12	0.02 ± 0.00	0.59 ± 0.09	0.75 ± 0.23	0.70 ± 0.15
C16:0 iso *	0.10 ± 0.01	0.02 ± 0.00	0.02 ± 0.00	0.02 ± 0.00	0.01 ± 0.00	0.15 ± 0.02	0.11 ± 0.01	0.10 ± 0.02	0.01 ± 0.00	0.00 ± 0.00	0.12 ± 0.01	0.10 ± 0.02
C16:0	18.36 ± 0.77	11.90 ± 0.34	12.54 ± 0.93	20.62 ± 1.22	11.82 ± 0.61	28.31 ± 0.66	13.36 ± 0.88	16.90 ± 1.01	10.75 ± 0.52	21.20 ± 1.01	25.85 ± 0.75	19.94 ± 0.57
C17:0	0.39 ± 0.05	0.25 ± 0.03	0.36 ± 0.09	1.13 ± 0.37	0.13 ± 0.01	1.25 ± 0.28	0.50 ± 0.13	0.10 ± 0.04	0.08 ± 0.01	1.32 ± 0.17	1.55 ± 0.17	0.55 ± 0.08
C18:0	5.13 ± 0.70	2.92 ± 0.11	4.52 ± 1.05	6.52 ± 1.02	3.77 ± 0.56	6.05 ± 0.81	2.67 ± 0.33	3.15 ± 0.46	1.85 ± 0.20	9.95 ± 0.91	12.73 ± 0.22	3.60 ± 0.28
C20:0	0.18 ± 0.03	0.29 ± 0.05	0.38 ± 0.06	0.39 ± 0.10	0.38 ± 0.07	0.21 ± 0.07	0.14 ± 0.02	0.12 ± 0.03	0.37 ± 0.07	0.51 ± 0.07	1.01 ± 0.37	0.26 ± 0.04
C22:0	0.12 ± 0.02	0.10 ± 0.0q	0.13 ± 0.02	0.24 ± 0.08	0.09 ± 0.01	0.26 ± 0.04	0.07 ± 0.01	0.10 ± 0.02	0.12 ± 0.02	0.26 ± 0.04	2.04 ± 0.49	0.42 ± 0.10
C24:0	0.06 ± 0.01	0.05 ± 0.00	0.07 ± 0.02	0.12 ± 0.03	0.06 ± 0.00	1.06 ± 0.22	0.05 ± 0.01	0.13 ± 0.02	0.18 ± 0.03	0.13 ± 0.01	0.84 ± 0.13	0.30 ± 0.07
∑ SFA	31.29 ± 1.45 ^AB^	19.46 ± 1.22 ^AB^	19.25 ± 1.20 ^AB^	34.70 ± 1.78 ^AB^	17.92 ± 1.02 ^AB^	42.78 ± 1.70^B^	22.77 ± 1.27 ^AB^	25.29 ± 1.55 ^AB^	13.55 ± 1.08^A^	36.57 ± 1.46 ^AB^	50.48 ± 1.77^B^	29.66 ± 1.33 ^AB^
C16:1 n-7	7.50 ± 1.10	2.01 ± 0.41	1.48 ± 0.43	3.67 ± 0.85	2.38 ± 0.55	1.75 ± 0.40	3.63 ± 0.77	2.83 ± 0.50	0.15 ± 0.01	3.77 ± 0.48	8.87 ± 1.01	3.73 ± 0.41
C17:1	0.06 ± 0.01	0.24 ± 0.04	0.18 ± 0.04	0.44 ± 0.08	0.11 ± 0.01	0.46 ± 0.15	0.54 ± 0.10	0.56 ± 0.12	0.04 ± 0.00	0.01 ± 0.03	1.32 ± 0.35	0.27 ± 0.05
C18:1 n-9	15.16 ± 1.11	40.35 ± 1.56	57.78 ± 3.66	16.07 ± 1.55	60.11 ± 2.01	11.06 ± 0.88	14.36 ± 0.79	14.45 ± 0.90	29.04 ± 1.07	24.53 ± 0.98	12.44 ± 0.73	21.93 ± 0.67
C18:1 n-7 *	4.25 ± 0.91	1.71 ± 0.22	2.15 ± 0.77	2.82 ± 0.68	2.15 ± 051	2.55 ± 0.44	1.89 ± 0.29	2.30 ± 0.44	0.58 ± 0.13	3.37 ± 0.40	5.18 ± 1.01	5.68 ± 0.38
C20:1 n-11 *	0.21 ± 0.02	0.31 ± 0.06	0.06 ± 0.02	0.17 ± 0.03	0.02 ± 0.00	0.07 ± 0.01	5.63 ± 0.79	4.44 ± 1.11	0.11 ± 0.01	0.74 ± 0.15	0.13 ± 0.02	0.11 ± 0.01
C20:1 n-7 *	0.43 ± 0.08	0.06 ± 0.01	0.08 ± 0.02	0.24 ± 0.07	0.02 ± 0.00	0.03 ± 0.00	0.68 ± 0.22	0.22 ± 0.08	0.22 ± 0.07	0.50 ± 0.09	0.12 ± 0.02	0.04 ± 0.00
C20:1 n-9	2.72 ± 0.55	4.63 ± 0.48	0.70 ± 0.07	2.42 ± 0.54	0.35 ± 0.09	0.49 ± 0.03	3.61 ± 0.45	2.69 ± 0.63	0.25 ± 0.06	0.78 ± 0.10	0.25 ± 0.10	1.26 ± 0.10
C22:1 n-11 *	1.65 ± 0.61	6.15 ± 0.95	0.27 ± 0.05	1.32 ± 0.33	0.04 ± 0.00	0.21 ± 0.01	10.89 ± 0.77	6.87 ± 1.09	0.02 ± 0.00	0.50 ± 0.06	0.07 ± 0.01	0.94 ± 0.08
C22:1 n-9	0.24 ± 0.02	0.37 ± 0.10	0.12 ± 0.03	0.52 ± 0.12	0.03 ± 0.00	0.11 ± 0.01	1.21 ± 0.27	0.96 ± 0.56	0.01 ± 0.00	0.13 ± 0.02	0.07 ± 0.01	0.19 ± 0.02
C24:1 n-9	0.50 ± 0.09	0.42 ± 0.15	0.33 ± 0.10	0.64 ± 0.21	0.14 ± 0.02	1.63 ± 0.52	1.11 ± 0.49	0.79 ± 0.32	0.04 ± 0.00	0.17 ± 0.03	0.51 ± 0.09	0.27 ± 0.04
∑ MUFA	33.20 ± 1.77 ^AB^	56.71 ± 2.25^B^	63.34 ± 4.11^C^	28.72 ± 1.65^A^	65.58 ± 2.67^C^	18.91 ± 0.86^A^	44.36 ± 2.11 ^AB^	37.08 ± 4.02 ^AB^	30.56 ± 1.62^A^	34.73 ± 1.23 ^AB^	29.40 ± 0.33^A^	35.01 ± 0.63 ^AB^
C16:2 n-4 *	1.26 ± 0.31	0.28 ± 0.02	0.18 ± 0.05	0.68 ± 0.13	0.21 ± 0.02	0.44 ± 0.06	0.20 ± 0.02	0.09 ± 0.01	0.04 ± 0.00	0.31 ± 0.04	0.20 ± 0.02	2.17 ± 0.18
C16:3 n-4 *	0.83 ± 0.25	0.07 ± 0.01	0.10 ± 0.02	0.36 ± 0.09	0.22 ± 0.03	0.03 ± 0.00	0.07 ± 0.01	0.33 ± 0.09	0.01 ± 0.00	0.34 ± 0.05	0.06 ± 0.01	0.03 ± 0.00
C16:4 n-4 *	1.60 ± 0.44	0.11 ± 0.01	0.04 ± 0.01	0.08 ± 0.02	0.53 ± 0.19	0.18 ± 0.02	0.14 ± 0.01	0.13 ± 0.03	0.02 ± 0.00	0.19 ± 0.02	0.45 ± 0.06	0.05 ± 0.00
C18:2 n-6	1.11 ± 0.41	5.33 ± 0.91	3.83 ± 0.72	1.47 ± 0.40	5.71 ± 0.80	1.72 ± 0.33	2.13 ± 0.19	1.30 ± 0.58	50.85 ± 3.55	7.16 ± 0.83	1.27 ± 0.22	4.86 ± 0.29
C18:3 n-3	0.56 ± 0.11	1.30 ± 0.31	0.56 ± 0.26	0.81 ± 0.15	0.57 ± 0.10	0.33 ± 0.06	1.16 ± 0.15	1.22 ± 0.77	0.86 ± 0.13	0.86 ± 0.20	0.30 ± 0.07	0.42 ± 0.14
C18:4 n-3 *	1.71 ± 0.22	2.94 ± 0.49	0.21 ± 0.10	0.96 ± 0.11	0.51 ± 0.13	0.31 ± 0.07	2.48 ± 0.33	2.07 ± 0.32	0.02 ± 0.00	0.43 ± 0.11	0.10 ± 0.01	0.31 ± 0.11
C20:2 n-6	0.13 ± 0.02	0.12 ± 0.02	0.08 ± 0.02	0.30 ± 0.03	0.02 ± 0.00	0.35 ± 0.10	0.50 ± 0.07	0.28 ± 0.08	0.02 ± 0.00	0.29 ± 0.03	0.26 ± 0.05	0.31 ± 0.10
C20:4 n-6	1.05 ± 0.30	0.24 ± 0.05	0.86 ± 0.22	2.27 ± 0.41	0.19 ± 0.02	1.20 ± 0.21	0.63 ± 0.13	0.46 ± 0.12	0.05 ± 0.01	2.62 ± 0.69	1.96 ± 0.34	1.60 ± 0.15
C20:4 n-3 *	0.72 ± 0.19	0.64 ± 0.19	0.11 ± 0.03	0.51 ± 0.19	0.16 ± 0.02	0.17 ± 0.02	1.42 ± 0.22	1.37 ± 0.56	0.02 ± 0.00	0.43 ± 0.15	0.72 ± 0.12	0.20 ± 0.02
C20:5 n-3	12.08 ± 0.80 ^B^	3.03 ± 044 ^A^	1.63 ± 0.34 ^A^	5.81 ± 0.79 ^A^	3.48 ± 0.62 ^A^	3.96 ± 0.22 ^A^	6.59 ± 0.98 ^A^	7.23 ± 1.33 ^A^	0.07 ± 0.01 ^A^	4.86 ± 0.47 ^A^	4.35 ± 0.78 ^A^	9.70 ± 0.51 ^B^
C21:5 n-3 *	0.66 ± 0.22	0.22 ± 0.05	0.20 ± 0.07	0.10 ± 0.02	0.14 ± 0.01	0.07 ± 0.01	0.37 ± 0.03	0.18 ± 0.04	0.29 ± 0.07	0.34 ± 0.07	0.05 ± 0.00	0.12 ± 0.01
C22:2	0.05 ± 0.01	0.02 ± 0.00	0.00 ± 0.00	0.04 ± 0.01	0.10 ± 0.01	0.04 ± 0.00	0.11 ± 0.01	0.04 ± 0.01	0.02 ± 0.00	0.01 ± 0.28	0.38 ± 0.09	0.92 ± 0.12
C22:5 n-6 *	0.20 ± 0.03	0.20 ± 0.06	0.34 ± 0.15	0.98 ± 0.30	0.03 ± 0.00	0.73 ± 0.21	0.17 ± 0.02	0.15 ± 0.03	0.04 ± 0.00	0.25 ± 0.04	0.25 ± 0.07	0.16 ± 0.04
C22:5 n-3 *	3.72 ± 0.67	0.56 ± 0.20	0.62 ± 0.21	2.08 ± 0.55	0.40 ± 0.13	0.55 ± 0.12	2.26 ± 0.34	1.81 ± 0.66	0.02 ± 0.00	0.70 ± 0.11	0.25 ± 0.05	0.81 ± 0.16
C22:6 n-3	6.48 ± 1.00 ^AB^	6.01 ± 1.55 ^AB^	5.34 ± 0.70	16.04 ± 1.88	1.35 ± 0.33 ^A^	25.09 ± 1.55 ^C^	11.41 ± 1.21 ^B^	16.39 ± 2.11	0.33 ± 0.10 ^A^	4.52 ± 0.63 ^AB^	4.07 ± 0.65 ^AB^	9.05 ± 0.39 ^AB^
∑ PUFA	33.10 ± 1.51 ^AB^	21.46 ± 2.78 ^AB^	14.28 ± 1.87 ^A^	33.19 ± 2.02 ^B^	13.84 ± 0.66 ^A^	35.46 ± 2.10 ^B^	30.44 ± 2.56 ^AB^	33.62 ± 2.54^B^	52.73 ± 2.88 ^C^	23.96 ± 1.59 ^AB^	15.16 ± 1.21 ^A^	31.08 ± 1.50 ^AB^
∑ n-3	26.01 ± 1.13 ^B^	14.83 ± 2.54 ^AB^	8.70 ± 1.33 ^AB^	26.46 ± 2.06 ^B^	6.62 ± 0.50 ^A^	30.57 ± 1.88 ^B^	25.92 ± 2.22 ^B^	30.44 ± 3.01 ^B^	1.62 ± 0.54 ^A^	12.15 ± 1.05 ^AB^	9.89 ± 0.92 ^AB^	20.69 ± 1.03 ^AB^
∑ n-6	2.73 ± 0.53 ^A^	6.04 ± 0.66 ^AB^	5.20 ± 1.04 ^AB^	5.35 ± 0.89 ^AB^	6.02 ± 0.76 ^AB^	4.07 ± 0.58 ^AB^	3.70 ± 0.24 ^AB^	2.39 ± 0.49 ^A^	50.99 ± 2.29 ^C^	10.52 ± 1.09 ^B^	3.85 ± 0.55^AB^	7.02 ± 0.35 ^AB^
n-6/n-3	0.10 ± 0.01 ^A^	0.41 ± 0.21 ^A^	0.60 ± 0.17 ^A^	0.20 ± 0.03 ^A^	0.91 ± 0.20 ^A^	0.13 ± 0.02 ^A^	0.14 ± 0.01 ^A^	0.08 ± 0.02 ^A^	31.48 ± 2.01 ^B^	0.87 ± 0.22 ^A^	0.39 ± 0.10^A^	0.34 ± 0.02 ^A^
Undefined	2.41 ± 0.36	2.37 ± 0.50	3.13 ± 0.54	3.39 ± 0.38	2.66 ± 0.43	2.85 ± 0.75	2.43 ± 0.55	4.01 ± 0.98	3.16 ± 0.79	4.74 ± 1.26	4.96 ± 0.47	4.25 ± 0.19
PUFA/SFA	1.06 ± 0.23 ^B^	1.10 ± 0.20 ^B^	0.74 ± 0.21 ^AB^	0.96 ± 0.15 ^B^	0.77 ± 0.15 ^AB^	0.83 ± 0.22 ^AB^	1.34 ± 0.41 ^B^	1.33 ± 0.40 ^B^	3.89 ± 0.87 ^C^	0.66 ± 0.16 ^AB^	0.30 ± 0.11^A^	1.04 ± 0.27 ^B^
EPA+DHA	18.56 ± 1.55 ^B^	9.04 ± 1.01 ^AB^	6.97 ± 1.05 ^AB^	21.85 ± 2.76 ^BC^	4.83 ± 0.87 ^A^	29.05 ± 2.56 ^C^	18.00 ± 1.49 ^B^	23.62 ± 3.22 ^BC^	0.40 ± 0.11 ^A^	9.38 ± 0.46 ^AB^	8.42 ± 0.89 ^AB^	18.75 ± 1.31 ^B^
DHA/EPA	0.54 ± 0.04 ^A^	1.98 ± 0.09 ^AB^	3.28 ± 0.59 ^B^	2.76 ± 0.32 ^AB^	0.39 ± 0.02 ^A^	6.34 ± 0.25 ^C^	1.73 ± 0.08 ^AB^	2.27 ± 0.71 ^AB^	4.71 ± 0.11 ^B^	0.93 ± 0.05 ^A^	0.94 ± 0.06 ^A^	0.93 ± 0.05 ^A^
AI	0.70 ± 0.15 ^AB^	0.34 ± 0.10 ^AB^	0.62 ± 0.12 ^AB^	0.21 ± 0.04^A^	0.25 ± 0.06 ^AB^	0.81 ± 0.21^B^	0.45 ± 0.09 ^AB^	0.46 ± 0.09 ^AB^	0.13 ± 0.03^A^	0.49 ± 0.13 ^AB^	1.04 ± 0.28 ^B^	0.51 ± 0.03 ^AB^
TI	0.28 ± 0.02 ^AB^	0.23 ± 0.03 ^AB^	0.29 ± 0.08 ^AB^	0.31 ± 0.05 ^AB^	0.30 ± 0.03 ^AB^	0.34 ± 0.02 ^AB^	0.19 ± 0.01 ^A^	0.19 ± 0.03 ^A^	0.28 ± 0.02 ^AB^	0.55 ± 0.02^B^	0.89 ± 0.02 ^B^	0.31 ± 0.02 ^AB^
FLQ	19.02 ± 0.79 ^B^	9.26 ± 1.07 ^AB^	7.20 ± 1.01 ^AB^	22.62 ± 1.55 ^B^	4.96 ± 0.38 ^AB^	29.90 ± 1.87 ^C^	18.45 ± 1.77 ^B^	24.61 ± 1.74 ^B^	0.41 ± 0.15 ^A^	9.84 ± 1.12 ^AB^	8.86 ± 1.16 ^AB^	19.58 ± 0.55 ^B^

**Table 3 foods-12-02631-t003:** Percentage of fatty acids (∑ SFA, ∑ MUFA, and ∑ PUFA) and quality indices in raw fish products as reported in the literature.

	Salmon	Mackerel	Sardine	Tuna	Shrimp	Crab
∑ SFA (%)	12.1 ± 0.8 (*Salmo salar*) [29] 19.0 ± 1.0–25.6 ± 2.9 (*Salmo salar*) [47] 13.6 ± 1.98–23.25 ± 1.93 (*Salmo salar*) [30] 19.2 [52]	44.6 ± 1.9 (*Scombers combrus)* [29] 30.1 ± 1.9 (*Pneumatophorus japonicus*) 31.4 ± 1.2 (*Scomberomorus maculatus*) [32] 27.6 [52]	54.2 ± 2.8 (*Sardina pilchardus*) [29] 50.8 [52]	45.5 ± 1.8 (*Thunnus thynnus*) [29] 53.8 [52]	33.2 ± 1.2 (*Parapenaeus longirostris*) [29] 25.6 ± 1.0 (*Oratosquilla oratoria*) 34.0 ± 1.2 (*Penaeus vannamei*) [32]	16.0 (*C. angulatus*) 18.3 (*C. japonicus*) [33] 22.58 ± 0.5–23.49 ± 0.3 (*C. mediterraneus*) [53]
∑ MUFA (%)	67.1 ± 1.4 (*Salmo salar*) [29] 33.4 ± 7.9–53.7 ± 3.9 (*Salmo salar*) [47] 32.73 ± 3.22 -52.89 ± 1.76 (*Salmo salar*) [30] 46.4 [52]	22.1 ± 3.0 *Scombers combrus)* [29] 29.6 ± 1.4 (*Pneumatophorus japonicus*) 33.3 ± 2.4 (*Scomberomorus maculatus*) [32] 29.8 [52] 26.13 ± 1.05 (*Trahurus mediterraneus*) [46]	22.1 ± 1.1 (*Sardina pilchardus*) [29] 26.6 [52]	30.5 ± 2.3 (*Thunnus thynnus*) [29] 20.6 [52]	22.9 ± 0.9 (*Parapenaeus longirostris*) [29] 22.3 ± 1.2 (*Penaeus vannamei*) 25.9 ± 1.3 (*Oratosquilla oratoria*) [32]	24.6 (*C. opilio*) 41.0 (*C. angulatus*) [33] 22.75 ± 0.5–24.01 ± 0.7 (*C. mediterraneus*) [53]
∑ PUFA (%)	27.3 ± 3.9–41.0 ± 5.8 (*Salmo salar*) [47] 29.93 ± 0.62–43.95 ± 2.77 (*Salmo salar*) [30] 29.9 [52]	30.4 ± 1.8 (*Pneumatophorus japonicus*) 33.8 ± 2.0 (*Scomberomorus maculatus*) [32] 35.9 [52] 35.04 ± 1.55 (*Trahurus mediterraneus*) [46]	23.7 [52]	24.4 [52]	38.8 ± 1.9 (*Penaeus vannamei*) 41.3 ± 1.3 *Oratosquilla oratoria*) [32]	35.3 (*C. angulatus*) 56.0 (*C. opilio*) [33] 36.12 ± 0.9–37.33 ± 0.9 (*C. mediterraneus*) [53]
PUFA/SFA	n.a.	0.90 ± 0.07 (*Trahurus mediterraneus*) [46]	n.a.	n.a.	n.a.	n.a.
EPA+DHA	10.1 [52]	28.3 [52]	13.4 [52]	5.8 [52]	n.a.	n.a.
AI	0.54 [52]	1.20 [52]	0.85 [52]	1.86 [52]	n.a.	n.a.
TI	0.21 [52]	0.17 [52]	0.59 [52]	0.74 [52]	n.a.	n.a.

n.a. = not available in the reference.

## Data Availability

The data used to support the findings of this study can be made available by the corresponding author upon request.

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
