# Peer review of "Determination of Fatty Acid Profile in Processed Fish and Shellfish Foods"

_foods, 2023, doi:10.3390/foods12132631_

Round 1
Reviewer 1 Report
Abstract: As an abstract should contain an introduction, material and methods, results and discussions, and conclusion, all in a well-summarized way, I believe that your abstract lacks a clearer conclusion of your results (as done in the simple summary from line 15 to line 18).
Introduction: nothing to comment
Materials and methods: nothing to comment
Results and discussions:
Page 8: The authors state that: “These results are comparable with those obtained by Mesa et al. [9], Grazina et al. [21] and Maldonado-Pereira et al. [22]”. It would be important to describe what was studied in these works that were used to compare with the results presented in the work. The example: These results are comparable with those obtained by Mesa et al. [9] in their work with Fresh Fishes samples, Shrimps, and Mollusks, Grazina et al. [21] in Salmon samples, and Maldonado-Pereira et al. [22] with ultra-processed foods.
Page 10, 3rd paragraph: I believe it would be important to discuss a little more about the results of the index, mainly comparing with the results of other works that evaluated the same types of foods studied.
Information that would be interesting for the reader would be to point out, even if in a table, results of works that indicate the percentage of fatty acids studied in fresh fish products so that it can be compared with the processed ones and thus infer about the effect of processing. Thus, the index results studied in fresh samples compared to processed ones.

Author Response
Reviewer 1
Comments and Suggestions for Authors
Abstract: As an abstract should contain an introduction, material and methods, results and discussions, and conclusion, all in a well-summarized way, I believe that your abstract lacks a clearer conclusion of your results (as done in the simple summary from line 15 to line 18).
R: The abstract has been modified.
Introduction: nothing to comment
Materials and methods: nothing to comment
Results and discussions:
Page 8: The authors state that: “These results are comparable with those obtained by Mesa et al. [9], Grazina et al. [21] and Maldonado-Pereira et al. [22]”. It would be important to describe what was studied in these works that were used to compare with the results presented in the work. The example: These results are comparable with those obtained by Mesa et al. [9] in their work with Fresh Fishes samples, Shrimps, and Mollusks, Grazina et al. [21] in Salmon samples, and Maldonado-Pereira et al. [22] with ultra-processed foods.
R: The changes have been made.
Page 10, 3rd paragraph: I believe it would be important to discuss a little more about the results of the index, mainly comparing with the results of other works that evaluated the same types of foods studied.
R: Comparisons with other studies have been included.
Information that would be interesting for the reader would be to point out, even if in a table, results of works that indicate the percentage of fatty acids studied in fresh fish products so that it can be compared with the processed ones and thus infer about the effect of processing. Thus, the index results studied in fresh samples compared to processed ones.
R: A table (Table 3) on the percentage of fatty acids in fresh fish products from studies in the literature was included.
Submission Date
23 April 2023
Date of this review
27 Apr 2023 23:26:26
Reviewer 2 Report
The MS “Determination of Fatty Acid Profile in Processed Fish and Shellfish Foods” of Vincenzo Nava, Vincenzo Lo Turco, Patrizia Licata, Veselina Panayotova, Katya Peycheva, Francesco Fazio, Rossana Rando, Giuseppa Di Bella and Angela Giorgia Potortì addresses the quality of several processed fish and shellfish foods according to their fatty acid composition.
General comments to the Authors.
The authors examined the FA composition (%) of several products, on the basis of which they calculated several indices (AI, TI, FLQ) and assessed the value of these products on human health. In the Introduction, in two paragraphs, the importance of the intake (quantity, mg/day) of omega-3 and omega-6 PUFAs is noted. Reading the manuscript, you expect to see these calculations and find out how much you need to consume these products to get a daily dose of omega-3 PUFAs. However, the authors did not measure these parameters in the studied products, which reduces the value of this study.
Based on the list of studied products, it is obvious that they were selected randomly and unstructured. There are 2 types of dried foods on this list, two types are natural, another is natural but grilled, and the main amount is canned foods, some of which additionally contained some kind of vegetable oil. In the case of the same species of animal in different types of processing, the authors compared how different types of processing affected the quality of the product. However, there are few such comparisons, which is due to the randomness of the choice of objects of study. It also reduces the value of the study.
The Discussion is poorly written. This topic is very popular: there are a large number of papers that show how canning and other processing types, and the addition of vegetable oils affect the content of PUFAs. Unfortunately, the authors do not provide comparisons with these studies. I have the impression that the authors found for the first time that the addition of vegetable oil leads to a strong change in the ratio of fatty acids.
Authors should take into account all these comments and improve the MS as much as possible.
Special comments.
Introduction
Line 46. The main source of n-6 PUFA is terrestrial agricultural animals like pork and chicken. Correct this sentence.
Line 58. The sentence is not correct. What does it mean "are not naturally". The genes of LC-PUFAs synthesis were found in human but the efficiency is not enough to supply the necessary quantity. Be more precise in scientific formulations.
Line 60. What guidelines? There are special organizations that make recommendations on dose consumptions. E.g., WHO...
Line 73. Change to “pro-inflammatory”
Table 1. Structure the table. First, fish, and fish of the same species together, then invertebrates, and also in groups. And do the same for table 2.
Line 74-75. Th reference (11) to a 30-year-old recommendation and the paper 4 with the same reference. Use up-to-date recommendations. Moreover, reference 11 lists PUFAs as a percentage of dietary energy.
Methods
Line 116-120. Provide more detail of the method. Acidic transmethylation is not a good method for fatty fish. Did you improve the method?
Line 152. What does “total lipids” mean here?
Results and Discussion
Line 161 and Table 2. Since the study is of applied importance, it is possible not to show minor fatty acids. For example, such FAs the percentage of which was above 0.5 or even 1%. I suggest to remove (4:0, 10:0, 11:0, 12:0, 13:0, 21:0, 23:0, 16:1n-9, 16:1n-5, 22:1n-7, 18:2n-4, 18:3n-6, 18:4n-1, 20:3n-6, and 20:3n-3).
Table 2. Are the “Undefined” peaks fatty acids? Why is the level of “uncertainties” in F4 so high (18%)?
Page 8. The first sentence. What explains this order of acids?
Page 8. Change “Saturated Fatty Acids” to “saturated fatty acids”
Page 8. paragraph 2, 3 and Page 10. paragraph 2, 3 These paragraphs duplicate data from tables. Remove the values, and instead add a comparison with the literature.
Page 8. “Good percentages” What does it mean?
Page 8. “considering the large contribution made by linoleic acid (50.85±3.55%).” It was already (a bit earlier) “of which linoleic acid (C18:2 n-6, 50.85±3.55%) made the greatest contribution”. It should be corrected.
Conclusion
The conclusion should summarize your results and highlight all the most important findings. Literature comparison should be removed. Instead, write that the FA composition depends on the quality of the raw product and the product processing techniques. Then, indicate which products are the most useful and which are less useful.
These sentences should be combined: “The most abundant fatty acids were DHA (C22:6 n-3), EPA (C20:5 n-3), palmitic (C16:0), stearic (C18:0), oleic (C18:1 n-9) and linoleic (C18:2 n-6) acids in all samples. The order of abundance was as follows: oleic acid (C18:1 n-9) > palmitic acid (C16:0) > linoleic acid (C18:2 n-6) > DHA (C22:6 n-3) > EPA (C20:5 n-3) > stearic acid (C18:0).”
Must be improved
Author Response
Reviewer 2
Comments and Suggestions for Authors
The MS “Determination of Fatty Acid Profile in Processed Fish and Shellfish Foods” of Vincenzo Nava, Vincenzo Lo Turco, Patrizia Licata, Veselina Panayotova, Katya Peycheva, Francesco Fazio, Rossana Rando, Giuseppa Di Bella and Angela Giorgia Potortì addresses the quality of several processed fish and shellfish foods according to their fatty acid composition.
General comments to the Authors.
The authors examined the FA composition (%) of several products, on the basis of which they calculated several indices (AI, TI, FLQ) and assessed the value of these products on human health. In the Introduction, in two paragraphs, the importance of the intake (quantity, mg/day) of omega-3 and omega-6 PUFAs is noted. Reading the manuscript, you expect to see these calculations and find out how much you need to consume these products to get a daily dose of omega-3 PUFAs. However, the authors did not measure these parameters in the studied products, which reduces the value of this study.
Based on the list of studied products, it is obvious that they were selected randomly and unstructured. There are 2 types of dried foods on this list, two types are natural, another is natural but grilled, and the main amount is canned foods, some of which additionally contained some kind of vegetable oil. In the case of the same species of animal in different types of processing, the authors compared how different types of processing affected the quality of the product. However, there are few such comparisons, which is due to the randomness of the choice of objects of study. It also reduces the value of the study.
The Discussion is poorly written. This topic is very popular: there are a large number of papers that show how canning and other processing types, and the addition of vegetable oils affect the content of PUFAs. Unfortunately, the authors do not provide comparisons with these studies. I have the impression that the authors found for the first time that the addition of vegetable oil leads to a strong change in the ratio of fatty acids.
Authors should take into account all these comments and improve the MS as much as possible.
Special comments.
Introduction
Line 46. The main source of n-6 PUFA is terrestrial agricultural animals like pork and chicken. Correct this sentence.
R: The correction has been made.
Line 58. The sentence is not correct. What does it mean "are not naturally". The genes of LC-PUFAs synthesis were found in human but the efficiency is not enough to supply the necessary quantity. Be more precise in scientific formulations.
R: The sentence has been modified.
Line 60. What guidelines? There are special organizations that make recommendations on dose consumptions. E.g., WHO...
R: The guidelines have been included.
Line 73. Change to “pro-inflammatory”
R: The correction has been made.
Table 1. Structure the table. First, fish, and fish of the same species together, then invertebrates, and also in groups. And do the same for table 2.
R: The tables have been changed.
Line 74-75. Th reference (11) to a 30-year-old recommendation and the paper 4 with the same reference. Use up-to-date recommendations. Moreover, reference 11 lists PUFAs as a percentage of dietary energy.
R: A more recent reference has been included.
Methods
Line 116-120. Provide more detail of the method. Acidic transmethylation is not a good method for fatty fish. Did you improve the method?
R: Yes, the method has been modified. In fact, to protect polyunsaturated fatty acids (PUFAs) from high temperatures, butylated hydroxytoluene (BHT) was used as an antioxidant. See line 125.
Line 152. What does “total lipids” mean here?
R: "Total lipids" has been changed to "total fatty acids," as also indicated in formula (3).
Results and Discussion
Line 161 and Table 2. Since the study is of applied importance, it is possible not to show minor fatty acids. For example, such FAs the percentage of which was above 0.5 or even 1%. I suggest to remove (4:0, 10:0, 11:0, 12:0, 13:0, 21:0, 23:0, 16:1n-9, 16:1n-5, 22:1n-7, 18:2n-4, 18:3n-6, 18:4n-1, 20:3n-6, and 20:3n-3).
R: Thanks for the advice, but we prefer to use the table as it stands.
Table 2. Are the “Undefined” peaks fatty acids? Why is the level of “uncertainties” in F4 so high (18%)?
R: The percentages were recalculated by not considering a significant unknown peak that contributed significantly to the percentage of undefined fatty acids. See Table 2.
Page 8. The first sentence. What explains this order of acids?
R: A explanation has been included.
Page 8. Change “Saturated Fatty Acids” to “saturated fatty acids”
R: The change has been made.
Page 8. paragraph 2, 3 and Page 10. paragraph 2, 3 These paragraphs duplicate data from tables. Remove the values, and instead add a comparison with the literature.
R: The changes have been made. A comparison has been included.
Page 8. “Good percentages” What does it mean?
R: The sentence has been changed.
Page 8. “considering the large contribution made by linoleic acid (50.85±3.55%).” It was already (a bit earlier) “of which linoleic acid (C18:2 n-6, 50.85±3.55%) made the greatest contribution”. It should be corrected.
R: The sentence has been corrected.
Conclusion
The conclusion should summarize your results and highlight all the most important findings. Literature comparison should be removed. Instead, write that the FA composition depends on the quality of the raw product and the product processing techniques. Then, indicate which products are the most useful and which are less useful.
R: The conclusions have been modified.
These sentences should be combined: “The most abundant fatty acids were DHA (C22:6 n-3), EPA (C20:5 n-3), palmitic (C16:0), stearic (C18:0), oleic (C18:1 n-9) and linoleic (C18:2 n-6) acids in all samples. The order of abundance was as follows: oleic acid (C18:1 n-9) > palmitic acid (C16:0) > linoleic acid (C18:2 n-6) > DHA (C22:6 n-3) > EPA (C20:5 n-3) > stearic acid (C18:0).”
R: The sentences were combined.
Comments on the Quality of English Language
Must be improved
Submission Date
23 April 2023
Date of this review
03 May 2023 14:41:31
Reviewer 3 Report
I have completed my review of the paper entitled " Determination of Fatty Acid Profile in Processed Fish and Shellfish Foods". The manuscript is well-written, and the text is easily understood. This work evaluates the fatty acid profiles of processed seafood and their nutritional quality using diverse indices. This research makes a valuable contribution to disclosing the fatty acids profiles of fish and shellfish, hence being of interest to those working in this area of knowledge. However, the manuscript can be improved, mainly in the results/discussion and conclusions (see the comments on the pdf of the manuscript).

Author Response
Reviewer 3
Comments and Suggestions for Authors
I have completed my review of the paper entitled " Determination of Fatty Acid Profile in Processed Fish and Shellfish Foods". The manuscript is well-written, and the text is easily understood. This work evaluates the fatty acid profiles of processed seafood and their nutritional quality using diverse indices. This research makes a valuable contribution to disclosing the fatty acids profiles of fish and shellfish, hence being of interest to those working in this area of knowledge. However, the manuscript can be improved, mainly in the results/discussion and conclusions (see the comments on the pdf of the manuscript).
Page 1, line 42: It would be better to rephrase this setence. As it is, seems that the highlighted part is one of the factors.
R: The change has been made
Page 2, line 45: Include abbreviation (i.e., PUFA) in the first time that is spelled. Therefore, delete the spelling 'polyunsaturated fatty acids' in the next two lines, keeping only "PUFA'
R: Changes have been made.
Page 2, lines 91-93: I think you should mention the evaluation performed through the nutritional indices
R: A mention on the assessment made through nutritional indices was included in the scope of the research.
Page 3, line 99-100: It would be better to clarify by adding some information, like "that assessed their Hg content".
R: The information has been included.
Table 1: Number of samples (i.e., No of Samples)
R: The change has been made.
Table 1: It should be specified (somewhere in the manuscript) the time of GC-FID analysis, in order to know if this were performed before or after the expiration date
R: This information was included in the text: p. 3, lines 103-104.
Page 3, line 103: purchased from Merck
R: The change has been made.
Page 5, line 157: Did you used any correction (e.g., Bonferroni)? This is recommended for multiple comparisons, as performed here
R: No correction was used.
Table 2: This legend must contain the correspondence between the sample code and the sample type "F1: Natural Shrimp, F2: ......", or at least a reference to Table 1 where this is contained.
R: A reference to Table 1 has been included.
Page 8, 2° paragraph: This and the next three paragraphs do not have any discussion. I think that would be interesting to add a brief discussion about the differences in the fatty acids classes found in the different sample types
R: A discussion has been included.
Page 8, 2° paragraph: Modify by (for example): "...acid (18:0), of which F8 had the highest percentage..."
R: The change has been made.
Page 8, 3° paragraph: respectively, followed by
R: The change has been made.
Page 8, 5° paragraph: Add a reference and compare with your results. Which sample types investigated are in line with these recommendations?
R: The information has been included.
Page 9, 1° paragraph: replace with "treatments as a possible..."
R: The change has been made.
Page 9, 2° paragraph: compared
R: The change has been made.
Page 9, 2° paragraph: delete ", in fact,"
R: The change has been made.
Page 9, 2° paragraph: samples
R: The change has been made.
Page 9, 2° paragraph: And the other fatty acid contents in these samples are higher or lower?
R: A comparison has been included.
Page 9, 2° paragraph: needs a reference
R: A reference has been included.
Page 9, 3° paragraph: profiles
R: The change has been made.
Page 9, 3° paragraph: The study listed as 26 studied the fatty acids composition of canned mackerel. Maybe you want to reference the study number 27. If so, be also aware of the reference numbering
R: The change has been made.
Page 9, 4° paragraph: You must give examples. Not all foods will promote the increment of MUFA and PUFA
R: Examples have been included.
Page 9, 4° paragraph: How? Add a simple discription of the differences observed between your study and Messias et al.
R: A description has been included.
Page 9, 4° paragraph: Delete, as it was not performed an analysis of variance for this comparison
R: The change has been made.
Page 10: Replace the two "in" with "than", and add some values of DHA obtained in your sampled to help readers
R: Changes have been made.
Page 10, 1° paragraph: fresh?
R: The change has been made.
Page 10, 1° paragraph: delete
R: The change has been made.
Page 10, 2° paragraph: give the values of FA classes, like you did for F1 in the previous sentence
R: The information has been included.
Page 10, 2° paragraph: Delete. This setence repeat the idea of the previous one
R: The change has been made.
Page 10, 2° paragraph: Replace with "of"
R: The change has been made.
Page 10, 2° paragraph: How different?
R: A comparison has been included.
Page 10, 3° paragraph: samples
R: The change has been made.
Page 10, 3° paragraph: needs a reference
R: A reference has been included.
Page 10, 3° paragraph: Therefore
R: The change has been made.
Page 10, 4° paragraph: Tuna pate, right? It is important to discriminate, because this value is very low to fish flesh. In fact, this should be further discussed. This low value likely results from the other foods added in the tuna pate.
R: The information has been included.
Conclusions: Conclusions must contain a clear about the contribution of this study for the field of seafood nutrition, as well provide the direction that future studies should follow
R: Changes have been made.
Conclusions: It would be better to rephrase this statement. For instance: "all samples tested proved to be a good source of the beneficial PUFA"
R: Changes have been made.
Comments on the Quality of English Language
Submission Date
23 April 2023
Date of this review
03 May 2023 10:58:57
Round 2
Reviewer 2 Report
The manuscript has been significantly improved, but it is necessary to justify the added data, since the calculations are not obvious.
The daily intake is expressed in absolute (for example in grams of omega-3 PUFAs per day) rather than relative content. A high percentage of omega-3 PUFAs does not mean a high concentration of these substances in the product. You haven't measured the absolute content so you can't conclude which products are better. Modify the sentence “Concerning the recommended daily intake for n-3 intake, among the analyzed samples F1, F4, F6, F7, F8 and F12 showed the better intake of Σ n-3.”
The authors added Table 4, which shows percentage intake of n-3 and n-6 PUFAs obtained from ingesting a portion (200g) of one of the processed products under investigation but neither the Methods nor the Results provide information on how these percentages were calculated. This information must be added. It is completely unclear how it was possible to calculate the % of PUFA consumption when consuming 200 g of the product in the absence of data on the amount of PUFA in products (mg/g of weight of the products).
The style can be improved.
Author Response
Comments and Suggestions for Authors
The manuscript has been significantly improved, but it is necessary to justify the added data, since the calculations are not obvious.
The daily intake is expressed in absolute (for example in grams of omega-3 PUFAs per day) rather than relative content. A high percentage of omega-3 PUFAs does not mean a high concentration of these substances in the product. You haven't measured the absolute content so you can't conclude which products are better. Modify the sentence “Concerning the recommended daily intake for n-3 intake, among the analyzed samples F1, F4, F6, F7, F8 and F12 showed the better intake of Σ n-3.”
R: The sentence has been modified.
The authors added Table 4, which shows percentage intake of n-3 and n-6 PUFAs obtained from ingesting a portion (200g) of one of the processed products under investigation but neither the Methods nor the Results provide information on how these percentages were calculated. This information must be added. It is completely unclear how it was possible to calculate the % of PUFA consumption when consuming 200 g of the product in the absence of data on the amount of PUFA in products (mg/g of weight of the products).
R: The formulas used to calculate the percentage were included in the manuscript.
Comments on the Quality of English Language
The style can be improved.
R: The style has been improved.
Round 3
Reviewer 2 Report
In the caption to Table 4 indicate the recommended values for the intake of n-3 and n-6 PUFA that you used in the calculation (equations 4 and 5), since this information was indicated only in the introduction.
You have added new data to Table 2. Specify the data source. If you have measured it, add it to Methods. If this is information from the manufacturer, then indicate it in the table.
Equations 4 and 5 have an error. According to equations 4, the result for n-3 F1 will be: 26 * 8.6 * 2 / 2 * 100 = 22360, not 223.6. So 100 should be removed. The same in Equations 5.
The sentence “The better percentage of ∑ n-3 PUFA were found in samples F1, F4, F6, F7, F8 and F12.” change to “The high percentage of ∑ n-3 PUFA were found in samples F1, F4, F6, F7, F8 and F12.”
I still recommend improving the style. The authors used words like "better", "our DHA/EPA ratios", and "our products" when they describe the data.
Author Response
DETAILED RESPONSE TO REVIEWERS
Reviewer 2
In the caption to Table 4 indicate the recommended values for the intake of n-3 and n-6 PUFA that you used in the calculation (equations 4 and 5), since this information was indicated only in the introduction.
- The authors added the recommended value of n-3 and n-6 PUFA in the caption to Table 4.
You have added new data to Table 2. Specify the data source. If you have measured it, add it to Methods. If this is information from the manufacturer, then indicate it in the table.
- The total lipid content of samples was experimentally determined by Folch method reported in section 2.3.
Equations 4 and 5 have an error. According to equations 4, the result for n-3 F1 will be: 26 * 8.6 * 2 / 2 * 100 = 22360,not 223.6. So 100 should be removed. The same in Equations 5.
- The equations 4 and 5 were revised.
The sentence “The better percentage of ∑ n-3 PUFA were found in samples F1, F4, F6, F7, F8 and F12.” change to “The high percentage of ∑ n-3 PUFA were found in samples F1, F4, F6, F7, F8 and F12.”
- The sentence was changed.
Comments on the Quality of English Language
I still recommend improving the style. The authors used words like "better", "our DHA/EPA ratios", and "our products" when they describe the data.
- The text was rewritten to avoid "better" and "our".